# Towards Evidence-Based Implementation of Pharmacogenomics in Southern Africa: Comorbidities and Polypharmacy Profiles across Diseases

**DOI:** 10.3390/jpm13081185

**Published:** 2023-07-26

**Authors:** Nyarai Desiree Soko, Sarudzai Muyambo, Michelle T. L. Dandara, Elizabeth Kampira, Dirk Blom, Erika S. W. Jones, Brian Rayner, Delva Shamley, Phumla Sinxadi, Collet Dandara

**Affiliations:** 1Platform for Pharmacogenomics Research and Translation (PREMED), University of Cape Town, South African Medical Research Council, Cape Town 7935, South Africa; 2Department of Pharmaceutical Technology, School of Allied Health Sciences, Harare Institute of Technology, Harare, Zimbabwe; 3Pharmacogenomics and Drug Metabolism Research Group, Division of Human Genetics, Department of Pathology and Institute of Infectious Diseases and Molecular Medicine, Faculty of Health Sciences, University of Cape Town, Cape Town 7935, South Africa; 4Department of Biological Sciences and Ecology, Faculty of Science, University of Zimbabwe, Harare, Zimbabwe; 5Medical Laboratory Sciences, School of Life Sciences and Health Professionals, Kamuzu University of Health Sciences (KUHES), Blantyre, Malawi; 6Division of Lipidology and Cape Heart Institute, Department of Medicine, Groote Schuur Hospital and Faculty of Health Sciences, University of Cape Town, Cape Town 7935, South Africa; 7Division of Nephrology and Hypertension, Department of Medicine, Groote Schuur Hospital and Faculty of Health Sciences, University of Cape Town, Cape Town 7935, South Africa; 8Division of Clinical Anatomy and Biological Anthropology, Department of Human Biology, Faculty of Health Sciences, University of Cape Town, Cape Town 7935, South Africa; 9Division of Clinical Pharmacology, Department of Medicine, Groote Schuur Hospital and Faculty of Health Sciences, University of Cape Town, Cape Town 7935, South Africa

**Keywords:** pharmacogenomics, translation, Africa, comorbidities, clinical

## Abstract

Pharmacogenomics may improve patient care by guiding drug selection and dosing; however, this requires prior knowledge of the pharmacogenomics of drugs commonly used in a specific setting. The aim of this study was to identify a preliminary set of pharmacogenetic variants important in Southern Africa. We describe comorbidities in 3997 patients from Malawi, South Africa, and Zimbabwe. These patient cohorts were included in pharmacogenomic studies of anticoagulation, dyslipidemia, hypertension, HIV and breast cancer. The 20 topmost prescribed drugs in this population were identified. Using the literature, a list of pharmacogenes vital in the response to the top 20 drugs was constructed leading to drug–gene pairs potentially informative in translation of pharmacogenomics. The most reported morbidity was hypertension (58.4%), making antihypertensives the most prescribed drugs, particularly amlodipine. Dyslipidemia occurred in 31.5% of the participants, and statins were the most frequently prescribed as cholesterol-lowering drugs. HIV was reported in 20.3% of the study participants, with lamivudine/stavudine/efavirenz being the most prescribed antiretroviral combination. Based on these data, pharmacogenes of immediate interest in Southern African populations include *ABCB1*, *CYP2B6*, *CYP2C9*, *CYP2C19*, *CYP2D6*, *CYP3A4*, *CYP3A5*, *SLC22A1*, *SLCO1B1* and *UGT1A1*. Variants in these genes are a good starting point for pharmacogenomic translation programs in Southern Africa.

## 1. Introduction

The main goal of pharmacogenomics is to improve patient care through the optimization of drug type and dosage selection, thus reducing the risk of adverse drug reactions and increasing patient adherence to treatment. Advances in genomics have greatly reduced the cost of genetic testing, and an ever-growing wealth of genomic evidence supports the integration of pharmacogenomics into clinical practice. In Western countries, pharmacogenomics is proving an effective addition to clinical decision making [1,2,3,4]. In Africa, however, pharmacogenomics is largely unutilized, due to the dearth of regional studies in genomics. Barriers to pharmacogenomics implementation in Africa [5] include the cost of genetic characterization, lack of genetics knowledge among healthcare workers, poorly resourced healthcare systems, socio-cultural and ethical issues, scientific and technical barriers and limited pharmacogenomic expertise and studies.

Southern Africa consists of 16 countries, with an estimated population of 360 million [6], predominantly Bantu-speaking inhabitants, and a median age of 22.1 years [6]. This relatively young population has a unique disease burden profile characterized by a high incidence of communicable diseases (e.g., HIV, tuberculosis (TB) and respiratory infections) and a rising burden of non-communicable diseases such as cancer and cardiovascular diseases (Figure 1). Communicable diseases like HIV, TB and malaria have long been the most important contributors [7] to the disease burden in Southern Africa, which has the highest HIV/AIDS prevalence globally and the highest worldwide morbidity due to HIV/AIDS (Figure 1). There are approximately 26 million people living with HIV (PLWH) and AIDS in Southern Africa [8]. Unfortunately, the non-communicable disease burden is rapidly growing in Southern Africa [7], driven predominantly by cardiovascular diseases and its associated risk factors, mental health disorders, neoplasms, diabetes and trauma.

For effective translation of pharmacogenomics in the region, it is important to map which drugs and combinations of drugs are frequently used in the population [9,10] and subsequently map the genetic variant profiles that interact with these drugs. In this study, we present data from multiple patient cohorts evaluating which drugs are commonly prescribed using four cohorts with participants on treatment for HIV/AIDS, hypertension, dyslipidemia and breast cancer. We provide motivation for the selection of pharmacogenes that may be important as a starting point towards developing pharmacogenomics tests as part of translating pharmacogenomics into clinical decision-making in Southern Africa.

## 2. Materials and Methods

This retrospective study is based on pooled data obtained from four cohorts with a total of 3997 patients recruited for pharmacogenomic studies in Southern African populations. Participants were recruited from these various studies from 2009 until 2022. The studies were all conducted in the Pharmacogenomics and Drug Metabolism Research Group, Division of Human Genetics, Department of Pathology, Faculty of Health Sciences, at the University of Cape Town. All participants who contributed blood or DNA samples to the biorepository consented to the use of their samples and data in pharmacogenetics/pharmacogenomics studies. All studies received ethical approval from the Human Research Ethics Committee (HREC) of the University of Cape Town (HREC ref: 231/2010). The four cohorts included: (i) a breast cancer cohort focusing on the pharmacogenetics of tamoxifen (*n* = 282; HREC ref: 501/2020); (ii) a cohort of patients with atrial fibrillation and mechanical valves on warfarin (*n* = 503; HREC ref: 581/2015); (iii) a cohort of patients recruited under the title “Pharmacogenomics of Cardiovascular Disease in South Africa” (PRECODE) (*n* = 2447; HREC ref: 694/2020), which focused on patients with hypertension (*n* = 1613) and patients with dyslipidemia (*n* = 834); and (iv) a cohort of patients recruited for studying the pharmacogenomics of antiretroviral therapy (*n* = 765; HREC ref: 104/2009).

For each cohort, the variables assessed included the drugs prescribed and the comorbidities recorded. Figure 2 shows the workflow of how both pharmacogenes and their variants relevant to Southern African populations were selected. Briefly, to identify the most frequent pharmacogenes relevant to the Southern African patient cohort, all drugs taken by each patient were counted. The top 20 drugs based on prescription frequency were then selected. To select pharmacogenes, we scouted the literature for the pharmacology of each drug and identified proteins, both transporters and enzymes, involved in the response to each drug. PharmGKB (www.pharmgkb.org accessed on 1 November 2022) [11] was used to confirm the listed genes associated with the response to each drug via the pharmacokinetic and pharmacodynamic pathways available in the PharmGKB database. Subsequently, each protein/enzyme involved in the response to the top drugs was noted. Pharmacogenetic variants per gene were subsequently selected based on both prior data from our research group (Table 1) and/or the literature and compared amongst global populations.

All of the listed studies (Table 1) employed participants from our biobank and patient database, including the 3997 patients described here. The participants recruited for the HIV/AIDS pharmacogenetics studies were recruited when a different profile of drug regimens was in use; thus, as dolutegravir (DTG) has currently replaced efavirenz as the first-line drug of choice for HIV, we also included genes involved in the metabolism of dolutegravir.

## 3. Results

We report a combined cohort consisting of 3997 patients. The demographics and clinical characteristics of the study participants have been described elsewhere [12,13,14,15,16,17,18,20,21,22]. The most common condition requiring treatment was hypertension (Figure 3), which was reported in 58.4% (2335/3997) of the patients. Consequently, antihypertensives were the most prescribed medications (Figure 3), with amlodipine, hydrochlorothiazide (HCT), enalapril and atenolol being the top four most prescribed medications in the combined cohort. At least 31.5% (1259/3997) of the study participants had dyslipidemia; hence, atorvastatin and simvastatin had a combined total prescription of 1483 patients (37.1%). At least 815 patients (20.3%) were reported to be living with HIV. The most common ARV combination prescription for PLWH was lamivudine/stavudine/efavirenz. The most prescribed antiretroviral (ARV) was lamivudine, prescribed for 717 (93.7%) of the 765 PLWH. The most prescribed anti-diabetic was metformin (676 patients), whilst tramadol (180 patients) was the most prescribed pain medication.

### 3.1. Prescriptions and Co-Prescriptions per Cohort

#### 3.1.1. Breast Cancer Cohort

The 282 women with breast cancer were recruited from Groote Schuur Hospital, Cape Town, South Africa, as part of a pharmacogenomic study of tamoxifen, and 54.2% (153/282) presented with at least one comorbidity. Patients without recorded comorbidities or co-prescribed drugs were excluded from further analysis. The most common comorbidity was hypertension in 41.2% (*n* = 63/153); followed by diabetes, which was reported among 34.6% (*n* = 53/153) of the patients; and bacterial infections which were reported among 26.8% (*n* = 41/153). The most commonly co-prescribed medications (Figure 4) were analgesics: paracetamol (78%) and tramadol (40%). The second most prescribed medications dealt with stomach and esophageal problems including Slow-Mag (25%), Senna (20%) and metoclopramide (6%). Antihypertensives as a group were the third most prescribed co-medication amongst women with breast cancer. Enalapril was the most prescribed antihypertensive drug, given to 19% of hypertensive breast cancer patients, followed by amlodipine (10%). Clonidine was prescribed to 13% of patients. Amoxicillin (7%) and flucloxacillin (8%) were the most commonly prescribed antibiotics in this cohort.

#### 3.1.2. Dyslipidemia Cohort

There were 834 patients in the pharmacogenomics of dyslipidemia study. Statins (Figure 4) were the major lipid-lowering drug used and were prescribed in 71.5% (603/843) of patients. Atorvastatin was the most prescribed statin (68.1%), whilst simvastatin was prescribed to 31% and rosuvastatin to 5% of the patients. Fluvastatin was prescribed to one patient, whilst pravastatin was prescribed to two patients. In total, 2% of study participants took bezafibrate to lower their triglyceride levels. Only two comorbidities were captured in the database of patients with dyslipidemia, namely hypertension, which occurred in 75% of patients, and diabetes which occurred in 25% of patients. Consequently, antihypertensives (HCT (20%), amlodipine (19%), atenolol (11%), enalapril (19%) and furosemide (3%)) and anti-diabetic agents (metformin (11%) and insulin (6%)) were the most co-prescribed medications.

#### 3.1.3. Hypertension Cohort

There were 1613 patients in the hypertension cohort archived in the Pharmacogenomics and Drug Metabolism database. The most prescribed antihypertensives (Figure 5) amongst these patients were amlodipine (89.8%; *n* = 1448/1613), enalapril (73.5%), HCT (71.7%), atenolol (54.9%) and furosemide (34.1%). Diabetes was the leading comorbidity in the hypertension cohort (Figure 5), occurring among 24% (*n* = 389/1613) of the patients, followed by dyslipidemia (16.2%; *n* = 261/1613), chronic kidney disease (15.6%; *n* = 253) and ischemic heart disease (11.3%; *n* = 183). Among hypertensive patients, co-medications were recorded in 24% of patients (*n* = 388/1613). The most prescribed anti-diabetic drugs were metformin, given to 79.1% of these patients (*n* = 307/1613); insulin, given to 39.4% (*n* = 153/1613); and gliclazide, given to 42.5% (*n* = 165/1613). The most prescribed lipid-lowering agent was simvastatin, in 34.3% of patients (*n* = 133/1613).

#### 3.1.4. Warfarin Cohort

There were 503 patients (453 South African and 154 Zimbabwean) patients enrolled in the pharmacogenomics of warfarin study in the biorepository. Amongst these patients, the most common comorbidities were hypertension (44%), heart failure (41%), dyslipidemia (21%), diabetes (12%) and HIV (9%). The most common co-prescribed medications (Figure 4) were statins (58% of patients) and antihypertensives (amlodipine (11%) and unspecified beta blockers (4%)), whilst efavirenz (14%) was the most commonly co-prescribed antiretroviral agent. At least 50% of the warfarin cohort [21] reported concomitant use of unspecified herbal supplements.

#### 3.1.5. HIV Cohort

A total of 765 PLWH were enrolled in the study on the pharmacogenomics of antiretrovirals within our biorepository. Of these, 616/765 (80.5%) were prescribed lamivudine/stavudine, with at least 310/616 (50.3%) on lamivudine/stavudine/efavirenz. Similarly, of the 75/765 (9.8%) patients on lamivudine/zidovudine, 26/75 (34.7%) were prescribed lamivudine/zidovudine/efavirenz. Efavirenz was present in 341/765 (44.6%) of prescriptions in this cohort of HIV patients, whilst lamivudine was present in 93.7% (717/765). The prophylactic antibiotic cotrimoxazole was the most commonly co-prescribed medication.

### 3.2. Selected Pharmacogenes and Their Variants

Amlodipine (Table 2) was the most prescribed drug (2020/3947 (51.2%). The response to amlodipine involves the genes CYP3A4, CYP3A5, ABCB1, ACE and CACNA1C. The protein angiotensin converting enzyme (ACE) is implicated in the response to antihypertensives (furosemide, HCT and amlodipine, enalapril), whilst the adrenergic receptor ADRB2 is implicated in response to three antihypertensives (spironolactone, atenolol and enalapril). DTG is metabolized by CYP3A4, UGT1A1, UGT1A3, UGT1A9, and its pharmacogenomic properties also involve the efflux pumps ABCG2 and ABCB1, as well as the influx pump SLC22A2. CYP3A4 metabolized 20% (Table 2) of the most prescribed drugs, whilst the influx pump SLC22A1 was implicated in the pharmacogenomics of 30% of the prescribed drugs.

The variants involved in the pharmacogenetics of the top 20 drugs prescribed in our cohort are also listed (Table 3). Variant frequencies in our population (Table 3) were extracted from previous work from our group [12,13,14,15,16,17,18,19,20,21,22,23]. Apart from the SLC22A1 variant rs34059508, which does not occur in Southern African populations, all variants listed in Table 3 could be variants of interest in the clinical utility of pharmacogenomics in Southern Africa.

## 4. Discussion

Pharmacogenomics is proving to be a useful tool in the delivery of safe and efficacious medications in many populations [1], but its application in the African clinical setting is still in its infancy. In order to translate evidence from the laboratory into the clinical setting, we set out to identify the main pharmacogenes and their variants that may be used in everyday clinical practice in Southern Africa. Using clinical data from patients in our study cohorts comprising individuals from Malawi, South Africa and Zimbabwe; we identified the most prescribed drugs and their respective pharmacogenes and variants of interest in the Southern African setting. Using the cohorts at our disposal, we report on the main disorders or diseases and provide a preliminary list of pharmacogenetic variants to be considered for translation to clinical utility in these populations. The list of variants in important pharmacogenes will continue to be updated as more information becomes available from ongoing studies.

The collision of disease burden between infectious and non-communicable diseases is seen through the cohorts we evaluated. The top ten causes of death in South Africa [25] are HIV/AIDS, ischemic heart disease, stroke, lower respiratory infections, diabetes, TB, road injuries, interpersonal violence, neonatal disorders and diarrheal disease. Zimbabwe has a similar picture to that of South Africa, with HIV/AIDS being the biggest threat among its approximately 16 million people [26], followed by lower respiratory infections, TB, ischemic heart disorders and neonatal disorders completing the top five causes of death. In Malawi, with a population of 18.6 million, maternal and neonatal disorders, HIV/AIDS, lower respiratory infections and TB and malaria are the top five causes of death [27]. Despite the inherent preponderance of cardiovascular disease in our study population, our study cohort represents the general ailments plaguing patients in Southern Africa; therefore, it provides a satisfactory basis to identify pharmacogenomic patterns in this population.

The epidemiological transition associated with progressive urbanization has led to an increase in cardiovascular risk factors like hypertension, diabetes, dyslipidemia and obesity [7]. These developments have resulted in an epidemiological shift that favors the rise of NCDs in Africa. NCDs accounted for 37% of deaths in Africa in 2019, up from 24% in 2000 [28]. Hypertension is an important risk factor for cardiovascular disease, contributing to the development of heart failure, atrial fibrillation, coronary artery disease, left ventricular hypertrophy, stroke, kidney failure and dementia [29,30]. At least 58% of our study participants were hypertensive, with 30% of participants outside the hypertension cohort reporting hypertension as a comorbidity. Amlodipine was the most prescribed antihypertensive, taken by approximately 90% of people with hypertension. Amlodipine should therefore be a top consideration for initiating pharmacogenomic testing in Southern Africa. Other antihypertensives of importance as evidenced by our cohort are enalapril, HCT, atenolol and furosemide.

Dyslipidemia, as measured by elevated cholesterol, is estimated to have a prevalence of 25.5% in African populations [31]. In our cohort, dyslipidemia was reported in 31.5% of the patients, whilst statins, in particular atorvastatin, were the lipid-lowering drugs of choice. The pharmacogenes involved in statin therapy should therefore be amongst the variants considered in the clinical utility of pharmacogenetics in Southern Africa. Similarly, the pharmacogenetics of the oral hypoglycemic metformin should also be considered in pharmacogenetic translation amongst African populations. There were 24 million people living with diabetes in Africa in 2021 [28,32], up from 19 million in 2019. The number of people living with diabetes in Africa is estimated to be 47 million in 2045 [28,32]. Diabetes will therefore require effective therapy aided by pharmacogenetic testing in clinical settings.

Although only 20.3% of the people in our study were reported to have HIV, HIV/AIDS remains the leading cause of morbidity and mortality in Southern African countries [25,26,27]. As this study included data from multiple cohorts selected for specific indications, the prevalence of HIV reported here is not representative of the true prevalence of HIV in the population. Efavirenz was the most prescribed ARV drug in South Africa, whilst lamivudine and stavudine were the most prescribed in Malawi. However, both cohorts were enrolled before the 2019 WHO recommendation [33] to prescribe DTG in combination with a nucleoside reverse transcriptase inhibitor as the preferred first-line therapy for PLWH instead of efavirenz. Zimbabwe [34], Malawi and South Africa [35] have all begun the process of changing over from efavirenz to dolutegravir as an important component of first-line therapy. Pharmacogenomic testing in HIV patients will therefore need to focus on DTG pharmacogenomics. DTG is provided in all three countries as a single-pill combination containing DTG, lamivudine and tenofovir [35]. Thus, the pharmacogenetics of all three of these drugs should be considered when selecting pharmacogenes of clinical importance in Southern Africa.

The drug transporter organic cationic transporter 1 (OCT1), which is encoded by the gene *SLC22A1*, potentially affects the response to at least 30% of the prescribed drugs in our cohort, including metformin and tramadol. The main polymorphisms implicated in the pharmacogenomics of OCT1 are rs12208357, rs34130495, rs72552763, rs628031 and rs34059508 [36,37], mostly resulting in an increased area under the curve for metformin. The *SLC22A1* SNP rs628031 is associated with the occurrence of adverse reactions to metformin [38] and, together with rs72552763, represents the only common SNP enough to be considered a priority variant in Southern Africa. *SLC22A2* encodes OCT2, which is the main facilitator of metformin uptake in the kidney [37]. OTC2 is also involved in the uptake of DTG. The most common *SLC22A2* variant is rs316019 (c.808G>T) which is linked to lactic acidosis [38], a potentially fatal reaction to metformin [(37) [24]. This variant occurs in at least 10% of all major global populations and can therefore be considered a priority variant for metformin pharmacogenomics.

Organic anion transporter protein B1 (OATPB1), which is encoded by *SLCO1B1,* is responsible for the hepatic uptake of statins and is a major pharmacogene in statin therapy. As expected, it is a pharmacogene of interest in atorvastatin and simvastatin therapy. The *SLCO1B1* variant rs4149056 (c.521T>C, p. Val174Ala) is implicated in statin-induced myopathy [39]. This variant results in reduced transporter activity that raises statin levels, including simvastatin [40,41], to plasma concentrations that result in myopathy. CPIC guidelines recommend testing for *SLCO1B1* c.521T>C when considering statin therapy [40]. Although this *SLCO1B1* variant is rare in African populations [23,40,42,43], it remains a priority variant in the multiracial Southern African population, particularly among the mixed-ancestry population. Similarly, efflux transporter *ABCG2* is also implicated in statin metabolism and is a CPIC-recommended pharmacogene of interest in statin therapy [44]. *ABCG2* rs2231142 (c.421C>A) results in the reduced activity of the efflux transporter and thus increased statin plasma levels and an increased risk of statin-induced myopathy [44]. *ABCG2* c.421A is also listed as an important variant in statin therapy by the CPIC [42]. However, even though the c.421A allele is virtually absent from African populations [23], it remains a priority variant among the mixed population groups. Certain African-specific variants in *SLCO1B1* and *ABCG2,* although occurring at high frequencies, lack enough evidence on their functional significance to warrant consideration in the pharmacogenetics of lipid-lowering therapy; thus, there is a call for more studies in individuals of African descent.

*ABCB1* encodes the efflux transporter multi-drug resistance protein 1/P-glycoprotein which is involved in the response to several of the drugs taken by patients in our combined cohort including amlodipine, simvastatin, atorvastatin, efavirenz, nevirapine, tramadol, warfarin and DTG. Despite its broad spectrum of substrates, only the *ABCB1* variant rs1045642 (c.3435C>T) has accumulated evidence of pharmacogenetic effects in disease management. The *ABCB1* c.3435T allele has been implicated in treatment outcomes of the antimalarials artemether and lumefantrine [45], mefloquine [46], warfarin stable-dose frequency [47] and breast cancer chemotherapy [48]. However, the c.3435T allele has varying frequencies in different populations, being low among Africans (0.19) and much higher among those with mixed ancestry (0.40) [(24) [11]. Despite this, *ABCB1* c.3435C>T remains a priority variant in the clinical translation of pharmacogenetics in Southern African populations, supported by the broad spectrum of important therapeutic agents that the transporter effluxes.

CYP3A4 metabolizes at least 20% of the most prescribed drugs mentioned in our study cohort, including amlodipine which was taken by 90% of the hypertensive patients. CYP3A4 is also responsible for DTG disposition. CYP3A4 and CYP3A5 share a high degree of sequence homology and considerable substrate overlap [49]; together they are involved in the response to 50–60% of drug substrates [50] and based on this evidence *CYP3A4* is marked a “very important pharmacogene” by PharmGKB [11]. There is substantial evidence of the involvement of CYP3A4*1B and CYP3A5*3, *6 and *7 in the pharmacogenomics of their substrates. CYP3A4*1B occurs in 82% of African individuals [51,52] and may therefore be an important variant in the clinical translation of pharmacogenetics in Southern Africa. *CYP3A5*3/*3* genotype is associated with complete absence of CYP3A5 activity [11]. CYP3A5*3 occurs at a frequency of 80–90% in individuals of European descent [43] and 66–96% in Asian populations; therefore, CYP3A5 activity is rare in these individuals. CYP3A5*3 is less frequent in individuals of African descent [11] and showed a marked variation in frequency between black Africans (0.19) when compared to South Africans of mixed ancestry (0.58) and may therefore be a priority variant in the clinical translation of pharmacogenetics in Southern Africa. Similarly, CYP3A5*6 and *7 are predominantly African variants [51,52] and almost absent in other populations; thus, they should be prioritized in pharmacogenetic testing in Southern African clinical settings.

CYP2B6 is an important drug-metabolizing enzyme involved in the response to both efavirenz and tamoxifen [53]. EFV was the backbone of first-line ART prior to the introduction of DTG. High plasma levels of EFV are linked to central nervous system toxicity. The *CYP2B6* c.516T/T genotype has been linked to increased EFV plasma levels [54,55,56] and, consequently, neurological toxicity. As the switch from EFV to DTG is still ongoing in most of Africa, *CYP2B6* c.516G>T (rs3745274) remains a variant of interest in African populations. This variant occurs in less than 30% and 20%, respectively, of Europeans and Asians, but has been reported at frequencies of at least 40% in Africans [55]. *CYP2B6* is also involved in the response to tamoxifen [53]; thus, it remains important regardless of the withdrawal of efavirenz. CYP2D6 metabolizes up to 30% of drugs commonly used in clinical practice and metabolizes tamoxifen and tramadol, which are both frequently used in the management of breast cancer in Southern Africa. Another gene of interest is *CYP2D6*17* (rs28371706, c.1023C>T), which occurs in at least 30% of individuals of African descent [57] and lowers the activity of the enzyme. This variant should be considered in the clinical utility of pharmacogenomics, especially in breast cancer patients. Also, we must consider the copy number variants of *CYP2D6* such as *2 × 2 which increases the function of the enzyme and can affect the drug metabolism of both tramadol and tamoxifen, especially in breast cancer patients within the region.

CYP2C9 is an important enzyme in the pharmacogenomics of cardiovascular diseases and therefore a potential priority pharmacogene in Africans in Southern Africa, where cardiovascular disorders are a growing burden. CYP2C9 is critical in warfarin therapy and simvastatin response. Both simvastatin and warfarin were among the top 20 prescribed drugs in the four cohorts. CYP2C9 is also involved in the metabolism of nonsteroidal anti-inflammatory drugs (NSAIDS) like ibuprofen and aspirin, both commonly used in pain management in our cohort. The CPIC has issued pharmacogenetic-based dose guidelines for both NSAIDS [58] and warfarin [2]. The most common polymorphisms in *CYP2C9* are *2 (R144C; rs1799853) and *3 (I359L; p rs1057910), which are both used in pharmacogenetic-based dosing guidelines for NSAIDS [58] and warfarin [2]. However, in African populations, CYP2C9*5, *6, *8 and *11 are more common [24] and therefore should be considered in the pharmacogenomics of CYP2C9 therapeutic substrates. CYP2C19 is mildly involved in the response to warfarin and tamoxifen. The loss of function variants *2 and *3 has been implicated in reduced enzyme activity. Homozygotes of *2/*2 and *3/*3 are classified as poor metabolizers [59]. Due to the minimal involvement of CYP2C19 as the primary metabolizing enzyme, this pharmacogene may not be of top priority for clinical application and can therefore be genotyped upon request.

Uridine diphosphate glucuronosyltransferases (UGT) proteins are a super family of enzymes responsible for the glucuronidation of target substrates like lipophilic drugs. UGT1A1 is involved in the disposition of DTG and paracetamol, a common analgesic as reported in the data on our combined cohort. The most common genetic variant affecting UGT1A1 function is the dinucleotide Tan repeat polymorphism (rs3064744) located in a TATAA consensus element [50]. The genotyping of rs3064744 allows for the detection of four main variations of this polymorphism, namely *UGT1A1*1* (TA6 reference genotype), *UGT1A1*28* (TA7 reference genotype), *UGT1A1*36* (TA5 reference genotype) and *UGT1A1*37* (TA8). *UGT1A1*28* and **37* both result in reduced enzyme activity, whilst *36 increases enzyme activity. A second *UGT1A1* polymorphism, *6 (rs4148323, c.211G>A; p.Gly71Arg) is also indicated in the drug response. The *UGT1A1* polymorphisms *28 and *6 result in reduced enzyme activity and thus affect the metabolism of the enzyme’s drug substrates [60]. Consequently, the FDA has issued a warning regarding the increased risk of the anticancer drug irinotecan which may induce neutropenia in individuals homozygous for the *28/*28 genotype [61]. UGT1A1*6 is rare in African populations and therefore may not be a priority polymorphism in our settings. However, *36 and *37 are exclusively African variants; thus, UGT1A1 rs3064744 may be a potentially important variant in pharmacogenetic clinical utility in Southern Africa.

A major limitation of this study is the exclusive use of records from our biorepository and the limited spread of data we obtained in relation to disease in Southern Africa. The pharmacogenomics research in our group has largely been driven by cardiovascular disease and HIV; thus, the data analyzed have a preponderance towards HIV and cardiovascular disorders. TB, malaria and mental health disorders play a significant role in the disease burden of Southern Africa and should also be considered in the clinical utility of pharmacogenomics in these populations.

## 5. Conclusions

The data reported from our four cohorts, taken together with knowledge from the literature and practice, give confidence on the morbidities, prescribed drugs and subsequent pharmacogenes that are of priority in the clinical utility of pharmacogenetics in Southern Africa. From our investigation, pharmacovariants of priority in our populations include variants in *SLCO1B1*, *SLC22A1*, *ABCB1*, *CYP2B6*, *CYP2D6*, *CYP2C9*, *CYP2C19* and *CYP3A4*. These pharmacovariants provide a good starting point for the clinical utility of pharmacogenomics in Southern Africa.

## Figures and Tables

**Figure 1 jpm-13-01185-f001:**
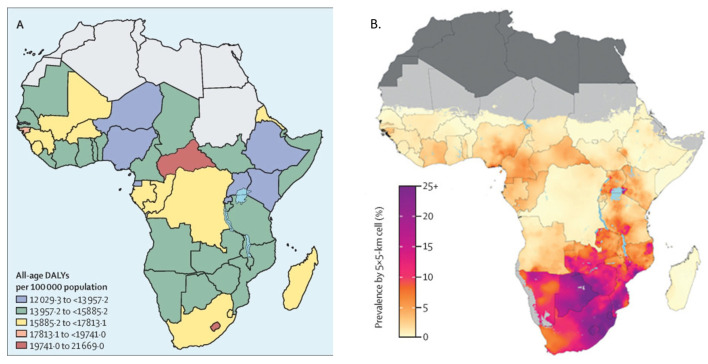
Disease burden in Southern Africa. (**A**) Non-communicable disease burden (reproduced with permission from [7]) and (**B**) HIV prevalence (reproduced with permission from [9]) in Africa in 2017.

**Figure 2 jpm-13-01185-f002:**
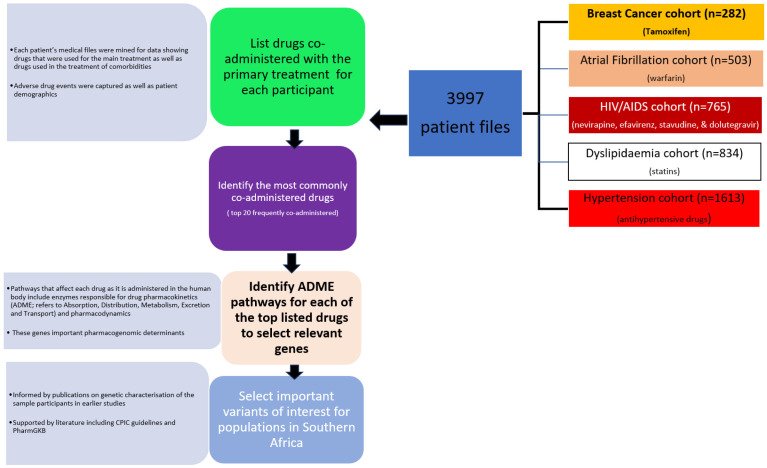
Flow chart of both pharmacogene and variant selection. Pharmacogenes were selected after enumeration of the top 20 most frequently co-administered drugs by prescription. Proteins involved in the pharmacodynamics and pharmacokinetics were noted and previous pharmacogenetic studies (Table 1) were employed to select variants.

**Figure 3 jpm-13-01185-f003:**
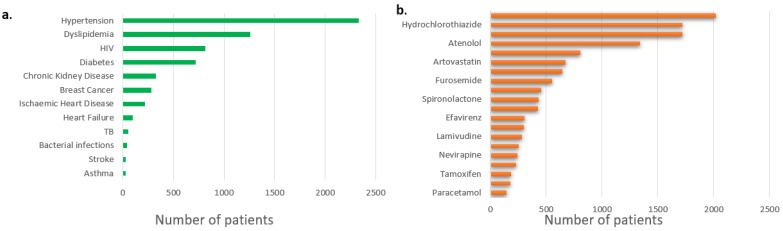
Summary of total population showing (**a**) number of patients reporting condition/disease and (**b**) top 10 most prescribed drugs in this population.

**Figure 4 jpm-13-01185-f004:**
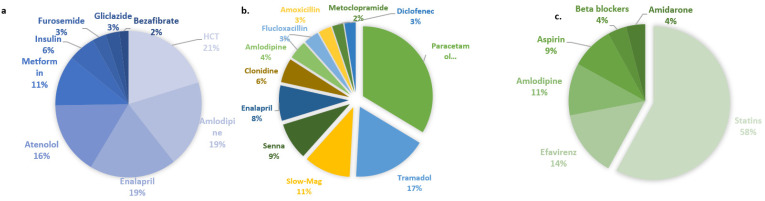
Co-prescriptions for patients in the (**a**) dyslipidemia cohort, (**b**) breast cancer cohort and (**c**) warfarin cohort. Pie charts show co-prescriptions for the dyslipidemia and breast cancer cohort, whilst for the warfarin cohort certain medications were recorded as classes only; for instance, statins and beta blockers.

**Figure 5 jpm-13-01185-f005:**
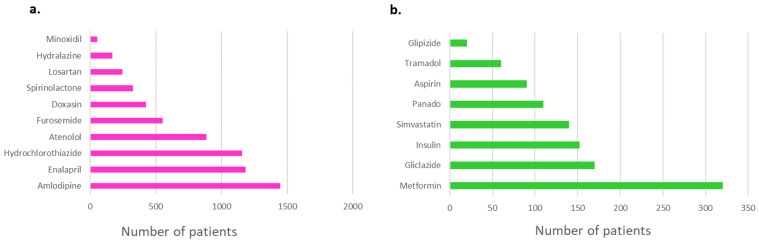
(**a**) Prescribed anti-hypertensives (*n* = 1613) and (**b**) co-prescribed medications (*n* = 388) of patients in the hypertension cohort. Of the 1613 patients in the hypertensive cohort, only 388 had recorded co-prescriptions, as shown in (**b**).

**Table 1 jpm-13-01185-t001:** List of studies that have reported on the four cohorts used in the current analysis showing genes that were characterized.

Cohort	Reference	*n*	Study Population	Phenotype Investigated	Pharmacogenes Genotyped (Investigated)
1. Pharmacogenomics of antiretroviral therapy	Swart et al., 2012 [12]	282	Bantu-speaking South Africans	Efavirenz plasma level, neuropsychological conditions, rashes, hallucination, lack of sleep	*ABCB1*
	Swart et al., 2012 [13]	464	Bantu-speaking South Africans	Efavirenz plasma level, neuropsychological conditions, rashes, hallucination, lack of sleep	*NR_1_I_3_*, *NR_1_I_2_*
	Swart et al., 2013 [14]	464	Bantu-speaking South Africans	Efavirenz plasma level, neuropsychological conditions, rashes, hallucination, lack of sleep	*CYP1A2*, *CYP2A6*, *CYP2B6*, *CYP3A4*, *CYP3A5*
	Kampira et al., 2014 [15]	203	Bantu-speaking Malawians	21% peripheral neuropathy 18% lipodystrophy 17% elevated lactate (>2.5 mMol/L)	*mtDNA:nDNA* ratio
	Mpeta et al.; 2016 [16]	86	Bantu-speaking South Africans	Lopinavir plasma concentration	*CYP3A4*
	Carr et al., 2017 [17]	151	Bantu-speaking Malawians	Nevirapine hypersensitivity	*GWAS*
	Carr et al., 2014 [18]	209	Bantu-speaking Malawians	Nevirapine hypersensitivity	*CYP2B6*
	Mhandire et al., 2015 [19]	118	Bantu-speaking Zimbabweans	Nevirapine hypersensitivity CD4 cell count	*CYP2B6*, *CYP1A2*
2. Atrial fibrillation and mechanical valves (warfarin)	Ndadza et al., 2021 [20]	503	Black and mixed-ancestry South Africans and Zimbabweans	Warfarin maintenance dose	*VKORC1*, *ABACB1*, *CYP2C9*, *CYP2C8*, *CYP1A2*, *CYP3A4*
	Muyambo et al., 2022 [21]	503	Black and mixed-ancestry South Africans and Zimbabweans	No phenotype investigated	29 pharmacogenes and 73 variants
3. Pharmacogenetics of tamoxifen	Unpublished data	282	Black, European and mixed-ancestry South Africans	Adverse drug reactions reported include hot flushes, pain, blood clots, depression, leg cramps, pins and needles, body aches, stroke, endometrial thickening, endometrial cancer, visual problems and recurrence	*CYP2D6*, *CYP3A4*, *CYP3A5*, *CYP2C9*, *CYP2C19*, *CYP2B6*, *SULT1A1*, *SULT1E1*, *SULT2A1*, *UGT1A4*, *UGT1A8*, *UGT1A10*, *UGT2B7*, *UGT2B15*
4. PRECODE—hypertension arm	Unpublished data	1613	Black and Mixed ancestry South Africans	Resistant hypertension, diabetes,	*ABCB1*, *CYP3A5*, *NEDD4L*, *SCNN1B*, *CES1*, *NR3C2*, *ADRB1*
5. PRECODE—dyslipidemia arm	Unpublished data	834	Black and Mixed ancestry South Africans	Statin associated myopathy, diabetes, hypertension	*ABCB1*, *ABCG2*, *CYP3A4*, *CYP3A5*, *SLCO1B1*
	Soko et al., 2016 [22]	30	Bantu speaking Zimbabweans	Rosuvastatin plasma levels	*ABCG2*, *SLCO1B1*
	Soko et al., 2018 [23]	30	Bantu speaking Zimbabweans	Rosuvastatin plasma levels	Whole exomes

Table 1 highlights pharmacogenes investigated for the different morbidities by our research group. This prior dataset informed the variants selected for clinical translation in Southern African populations.

**Table 2 jpm-13-01185-t002:** Pharmacogenes associated with the top 20 prescribed drugs.

*Drug*	*Pharmacogene*
*Amlodipine*	*CYP3A4 CYP3A5 CACNAIC ABCB1 ACE*
*Hydrochlorothiazide*	*ADD1 NEDD4L KCNJ1 WNK1 ACE*
*Enalapril*	*CES1 ACE VEGFA ABO ADRB2*
*Atenolol*	*ADRB2 ADRB1 AGT GNB3 GRK4*
*Simvastatin*	*SLCO1B1 ABCB1 CYP3A4 CYP3A5 CYP2C9 ABCG2*
*Atorvastatin*	*SLCO1B1 ABCB1 CYP3A5 APOE CYP3A4 ABCG2*
*Metformin*	*SLC22A1 SLC47A1 SLC47A2 SLC22A2 ATM*
*Furosemide*	*NPPA-ASI ACE ADD1 SCNN1G SLC12A3*
*Warfarin*	*CYP2C9 VKORC1 CYP4F2 GGCX CYP2C19*
*Spironolactone*	*ACE CYP4A11 ADRB1 ADRB2 ADD1*
*Doxazosin*	*ADRA1B ADRA2A KCNH2*
*Efavirenz*	*CYP2B6 NRI13 NRI12 UGT2B7 CYP2A6 ABCB1*
*Insulin*	*G6PD SCNN1B SLC30A8*
*Lamivudine*	*SLC22A1 OCT2*
*Gliclazide*	*KCNJ11 CYP2C9 ABCC8 KCNQ1*
*Nevirapine*	*CYP3A4 CYP2D6 CY2B6 ABCB1*
*Stavudine*	*Discontinued*
*Tamoxifen*	*CYP2D6 CYP2C19 CYP3A5 ABCC2 CYP2B6*
*Tramadol*	*CYP2D6 ABCB1 OPRM1 COMT SLC22A1*
*Panado*	*SULT1A1 SULT1A3 UGT1A1*
*Dolutegravir*	*CYP3A4 ABCG2 ABCB1 UGT1A1 UGT1A3 UGT1A9 SLC22A2*

**Table 3 jpm-13-01185-t003:** Variants of interest to Southern African populations associated with the top 20 prescribed drugs.

		The following polymorphism frequencies were obtained from Muyambo et al. [21]						
		Variant Allele Frequencies						
Gene	dbSNP	Africans (Southern Africa)	Mixed Ancestry (Southern Africa)	West African (YRI)	East African (LWK)	African Americans	East Africans	Europeans
*ABCB1*	rs1045642	0.09	0.40	0.13	0.14	0.23	0.40	0.52
*CYP2B6*	rs28399499	0.10	0.03	0.12	0.06	0.07	0.00	0.00
	rs3745274 (c.516G>T)	0.35	0.32	0.40	0.36	0.35	0.22	0.24
*CYP2C9*	rs1799853 (*2)	0.01	0.04	0.00	0.00	0.07	0.001	0.12
	rs1057910 (*3)	0.00	0.05	0.00	0.00	0.02	0.03	0.07
	rs7900194 (*8)	0.11	0.02	0.05	0.07	0.02	0.00	0.02
	rs2256871 (*9)	0.58	-----	0.09	0.15	0.07	0.00	0.001
*CYP2C19*	rs12248560 (*17)	0.14	0.13	0.25	0.18	0.22	0.02	0.22
	rs4244285(*2)	0.17	0.22	0.17	0.21	0.18	0.31	0.15
	rs28399504(*3)	1.00	0.00					
*CYP2D6*	rs1065852 (*10)	0.07	0.10	0.11	0.04	0.19	0.57	0.20
	rs72549357 (*15)	0.05	0.03	-----	-----	-----	-----	-----
	rs28371706 (*82)	0.19	0.04	0.26	0.19	0.14	0.00	0.00
	rs59421388 (*29)	0.15	0.006	0.11	0.17	0.07	0.00	0.00
	rs3892097 (*4)	0.02	0.11	0.06	0.03	0.15	0.00	0.19
	rs28371725 (*41)	0.03	0.05	0.09	0.03	0.09	0.04	0.09
	rs16947 (*34)	0.12	0.23	0.56	0.65	0.46	0.14	0.34
*CYP3A4*	rs35599367(*22)	0.00	0.03	0.00	0.00	0.00	0.00	0.005
*CYP3A5*	rs776746 (*3)	0.15	0.58	0.17	0.12	0.31	0.71	0.95
	rs10264272 (*6)	0.24	0.05	0.17	0.24	0.12	0.00	0.0003
	rs41303343 (*7)	0.14	0.04	0.12	0.12	0.04	0.00	0.00
*SLCO1B1*	rs4149056	0.005	0.08	0.009	0.02	0.04	0.12	0.16
	**The following polymorphism frequencies were obtained from 1000 Genomes** [24]							
**Gene**	**Variant ID**	**African**	**East Asian**	**European**	**South Asian**	**American**		
*CYP2B6*	rs4803419	0.08	0.44	0.32	0.34	0.35		
	rs12208357	0.004	0.00	0.06	0.02	0.02		
	rs34130495	0.003	0.00	0.02	0.01	0.07		
*SLC22A1*	rs72552763	0.05	0.005	0.18	0.15	0.29		
	rs34059508	0.00	0.00	0.02	0.00	0.02		
	rs628031	0.73	0.74	0.59	0.61	0.78		
*SLC22A2*	rs316019	0.19	0.14	0.11	0.13	0.09		
*UGT1A1*	rs4148323 (*6)	0.001	0.14	0.001	0.002	0.01		

## Data Availability

The data presented in this study are available on request from the corresponding author. The data are not publicly available due to privacy reasons.

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
