# Peer review of "Towards Evidence-Based Implementation of Pharmacogenomics in Southern Africa: Comorbidities and Polypharmacy Profiles across Diseases"

_jpm, 2023, doi:10.3390/jpm13081185_

Round 1
Reviewer 1 Report
The authors conducted the study using available data on 3997 individuals with an aim to identify pharmacogenetic variants for drugs used for various conditions in patients from Southern Africa. The article lacks novelty and several useful information. Following questions to the authors:
1. How were the variants selected for the various conditions? Did the authors looks at the current GWAS data? Were the selected variants reported in this article studied for their association with drug response in South African population?
2. How were the patients genotyped? The method section lacks information on genotyping platform used in the study.
3. How do allele frequency difference and LD structure affect pharmacogenomics within South Africa? There is population substructure within the Bantu speaking people in South Africa. However, the authors haven't provided such information or reference to previous pharmacogenomic studies in South Africa.
4. Were their any drug adverse effects or poor drug response in the 3997 patients?
Author Response
Comments and Suggestions for Authors
The authors conducted the study using available data on 3997 individuals  with an aim to identify pharmacogenetic variants for drugs used for various conditions in patients from Southern Africa. The article lacks novelty and several useful information.
Our response; This article serves to help design and implement pharmacogenomic testing at clinical level. Tools and pipelines are being developed to help in this regard. In our opinion; clinical implementation of pharmacogenetics in our communities is minimal and close to non existent driven primarily by lack of data hence the motivation of this article.
Following questions to the authors:
- How were the variants selected for the various conditions? Did the authors looks at the current GWAS data? Were the selected variants reported in this article studied for their association with drug response in South African population?
Our response: The 3997 participants in this manuscript were genotyped for several pharmacogenes as part of four different projects which culminated in the publication of several manuscripts (see Table 1). The variants were selected based on their potential role in the metabolism and disposition of the main drugs used in the treatment of the focus condition in each cohort from three countries in Malawi, South Africa and Zimbabwe. This manuscript, however, goes further to evaluate the possible drug-drug and drug-gene interactions, to inform pharmacogenomics clinical implementation. We have amended the methods section to explain how variants were selected. We have added a flow chart of our method as Figure 1.0 and have added Table 1.0 to allow reference of prior work we have done in genotyping these cohorts. The method section lines 109-121 now reads “For each cohort, variables assessed included drugs prescribed and comorbidities recorded. Figure 2 shows the workflow of how both pharmacogenes and their variants relevant to Southern African populations were selected. Briefly, to identify the most frequent pharmacogenes relevant to the Southern African patient cohort, all drugs taken by each patient were counted. The top 20 drugs based on prescription frequency were then selected. To select pharmacogenes, we scouted literature for the pharmacology of each drug and identified proteins both transporters and enzymes involved in the disposition of each drug. PharmGKB (www.pharmgkb.org) [11] was used to con-firm the listed genes associated with response to each drug via the pharmacokinetic and pharmacodynamic pathways available in the PharmGKB database. Subsequently, each protein/ enzyme involved in the disposition of the top drugs was noted. Pharmacogenetic variants per gene were subsequently selected based on both prior data from our research group (Table 1), and/ or literature and compared amongst global populations.
Figure 2 Flow chart of both pharmacogene and variant selection. Pharmacognosy were selected after enumeration of the top 20 most frequent co-administered drugs by prescription. Proteins involved in the pharmacodynamics and pharmacokinetics were noted and previous pharmacogenetic studies (Table1 ) were employed to select variants.
All listed studies (Table 1.0) employed participants from our biobank and patient database; including the 3997 described here.
- How were the patients genotyped? The method section lacks information on genotyping platform used in the study.
Our response:
The patients constitute participants enrolled in four pharmacogenomics studies and was carried out using methods such as PCR-RFLP, TaqMan, whole exome sequencing, genotyping Arrays, all confirmed by Sanger sequencing. Thus, the genetic characterisation results are not part of this manuscript BUT part of the various manuscripts as indicated in Table 1, as well some unpublished results. We have amended the methods section to include Table 1 which outlines genotyping of patients DNA samples.
- How do allele frequency difference and LD structure affect pharmacogenomics within South Africa?  There is population substructure within the Bantu speaking people in South Africa. However, the authors haven't provided such information or reference to previous pharmacogenomic studies in South Africa. 
Our response:
This is an important question. African populations are known to be genetically diverse, thus, the information of populations sampled in this part of Africa, is not going to be representative of all African populations but make a contribution to the understanding of such diversity. There are allele frequency and LD structure differences which further implores for more studies across the African continent to understand the spectrum of important pharmacogenes as they interact with the different geographical or environmental exposures.
. 4. Were their any drug adverse effects or poor drug response in the 3997 patients?
Our response: This is part of the focus of this paper. In the four cohorts, drugs used in the treatment of the main disease and those used to treat comorbidities were noted. Adverse drug effects were recorded as well. We have captured these ADRs in Table 1 below.
|
Cohort |
Reference |
n |
Study population |
Phenotype investigated |
Pharmacogenes genotyped (investigated) |
|
1. Pharmacogenomics of antiretroviral therapy |
Swart et al., 2012 |
282 |
Bantu speaking South Africans |
Efavirenz plasma level, neuropsychological conditions, rashes, hallucination, lack of sleep |
ABCB1 |
|
Swart et al., 2012 |
464 |
Bantu speaking South Africans |
Efavirenz plasma level, neuropsychological conditions, rashes, hallucination, lack of sleep
|
NR1I3, NR1I2 |
|
|
Swart et al., 2013 |
464 |
Bantu speaking South Africans |
Efavirenz plasma level, neuropsychological conditions, rashes, hallucination, lack of sleep
|
CYP1A2, CYP2A6, CYP2B6, CYP3A4, CYP3A5 |
|
|
Kampira et al., 2014 |
203 |
Bantu speaking Malawians |
21% peripheral neuropathy 18% lipodystrophy 17% elevated lactate (.2.5mMol/L) |
mtDNA:nDNA ratio |
|
|
Mpeta et al.; 2016 |
86 |
Banti speaking South Africans |
Lopinavir plasma concentration |
CYP3A4 |
|
|
Carr et al., 2017 |
151 |
Bantu speaking Malawians |
Nevirapine hypersensitivity |
GWAS |
|
|
Carr et al., 2014 |
209 |
Bantu speaking Malawians |
Nevirapine hypersensitivity |
CYP2B6 |
|
|
Mhandire et al., 2015 |
118 |
Bantu speaking Zimbabweans |
Nevirapine hypersensitivity CD4 cell count |
CYP2B6, CYP1A2 |
|
|
2. Atrial fibrillation and mechanical valves (warfarin) |
Ndadza et al., 2021 |
503 |
Black and Mixed ancestry South Africans and Zimbabweans |
Warfarin maintenance dose |
VKORC1, ABACB1, CYP2C9, CYP2C8, CYP1A2, CYP3A4 |
|
Muyambo et al., 2022 |
503 |
Black and Mixed ancestry South Africans and Zimbabweans |
No phenotype investigated |
29 pharmacogenes and 73 variants
|
|
|
3. Pharmacogenetics of tamoxifen |
Unpublished data |
282 |
Black, European and Mixed ancestry South Africans |
Adverse drug reactions reported include hot flushes, pain, blood clots, depression, leg cramps, pins and needles, body aches, stroke, endometrial thickening, endometrial cancer, visual problems and recurrence |
CYP2D6, CYP3A4, CYP3A5, CYP2C9, CYP2C19, CYP2B6, SULT1A1, SULT1E1, SULT2A1, UGT1A4, UGT1A8, UGT1A10, UGT2B7, UGT2B15 |
|
4. PRECODE-hypertension arm |
Unpublished data |
1613 |
Black and Mixed ancestry South Africans |
Resistant hypertension, diabetes, |
ABCB1, CYP3A5, NEDD4L, SCNN1B, CES1, NR3C2, ADRB1 |
|
5. PRECODE-dyslipidemia arm |
Unpublished data |
834 |
Black and Mixed ancestry South Africans |
Statin associated myopathy, diabetes, hypertension |
ABCB1, ABCG2, CYP3A4, CYP3A5, SLCO1B1 |
|
Soko et al., 2016 |
30 |
Bantu speaking Zimbabweans |
Rosuvastatin plasma levels |
ABCG2, SLCO1B1 |
|
|
Soko et al., 2018 |
30 |
Bantu speaking Zimbabweans |
Rosuvastatin plasma levels |
Whole exomes |
Reviewer 2 Report
Soko and colleagues aimed to explore the clinical implications of pharmacogenomics in four study cohorts of common diseases in South Africa. The clinical significance and timeliness of this research question cannot be overstated, as the scarcity of studies on PGx in African and non-European populations has impeded the widespread applicability and utility of PGx research to diverse populations across the globe. While the topic is of great importance, it is worth noting that the study’s omission of directly examining the impact of individuals' genotypes on the patterns of concomitant medication use and future prescribing. There are several major flaws in the manuscript and study design that need to be addressed before the findings can be considered reliable and meaningful. The following points outline the key concerns:
#1. The Results section appears disorganized and lacks consistency with Figure illustrations, specifically Figures 3 and 4. This inconsistency has posed challenges for me in evaluating the validity of the findings. Please confirm whether the findings and statistics are consistent between the corresponding sub-sections under Results and Figures. As one example to illustrate this issue among various scenarios:
Line 146-147: “the most common comorbidity (Figure 3) was hypertension…” The current Figure 3 summarizes co-prescriptions, not comorbidity.
#2. Caution should be exercised when generalizing the study findings to the broader population in South Africa, as the participants were selected from four study cohorts consisting of individuals with well-defined cases of specific diseases. The prevalence of these diseases and medication usage may not accurately reflect that of the general population in South Africa.
#3. More information is required regarding the curation process for drug-gene pairs, including the specific resources utilized and the level of evidence considered. Additionally, it is unclear whether the level of clinical actionability was assessed or evaluated in the study. In particular:
-
Line 111-113: “the author noted that PharmGKB and literature was used to identify genes associated with response to each drug.” It was unclear how was response defined in this context (e.g., efficacy, tolerability, or both?).
-
It is important to note that PharmGKB assigns levels of evidence to the annotation of drug-gene pairs. Those with low levels of evidence (e.g., 3 and 4) may not be clinically actionable. The drug-gene pairs included in the evidence-based peer-reviewed guidelines published by CPIC, DPWG, or other professional societies could be informative guidance in addition to PharmGKB.
-
It is also helpful to mention the date of data curation as the database can be periodically updated. For example, the PGx annotations of Amlodipine on PharmGKB at the time of this review are slightly different from what was included in this submission (https://www.pharmgkb.org/chemical/PA448388/clinicalAnnotation)
#4. Table 2. The rationale for the selection of the variants and their clinical actionability is also unclear to me. For example, some important haplotypes of CYP2C19 are missing, such as *2, *3, *9, *12, *14, *17, etc. Duplication and deletion in CYP2D6 are also missing but very important to consider.
Minor:
#5. To improve the readability, summarizing the clinical and demographic characteristics of study participants in a table could be helpful to improve the readability.
#6. Figure 2 - Confirm the completeness of the top 20 prescribed drugs (study cohort vs population). Some medications mentioned in the text (e.g., lines 132-133: metformin and tramadol) were not shown in the bar plot.
Quality of English language is fine.
Author Response
The Results section appears disorganized and lacks consistency with Figure illustrations, specifically Figures 3 and 4. This inconsistency has posed challenges for me in evaluating the validity of the findings. Please confirm whether the findings and statistics are consistent between the corresponding sub-sections under Results and Figures. As one example to illustrate this issue among various scenarios:
Line 146-147: “the most common comorbidity (Figure 3) was hypertension…” The current Figure 3 summarizes co-prescriptions, not comorbidity.
Our response: Results referring to Figures 2,3 and 4 revisited. Line 146-147 (now line 170), Figure 3 has been removed and this line now reads “The most common comorbidity was hypertension in 41.2% (n=63/153); followed by diabetes, “
#2. Caution should be exercised when generalizing the study findings to the broader population in South Africa, as the participants were selected from four study cohorts consisting of individuals with well-defined cases of specific diseases. The prevalence of these diseases and medication usage may not accurately reflect that of the general population in South Africa.
Our response: This study involves populations from three Africa countries Malawi, Zimbabwe, and South Africa (Line 239-240) and as highlighted by the reviewer caution has been taken not to over generalise our findings to the entire Southern African population. Lines 270-280 confirm the author’s awareness of the preponderance of the cohorts examined to primarily cardiovascular disease and HIV; however, the top diseases in the three countries under study currently are HIV/AIDS (which we tackle), ischemic heart disease and stroke (which we also tackle), TB (which was a co-morbidity in our participants), lower respiratory infections, diabetes (another major co-morbidity in our participants), malaria and neonatal disorders. Line 270-281 in particular read “Despite the inherent preponderance of cardiovascular disease of our study population, our study cohort represents the general ailments plaguing patients in the southern part of Africa; and therefore, provides a satisfactory basis to identify predict important pharmacogenomic patterns in this population.”
#3. More information is required regarding the curation process for drug-gene pairs, including the specific resources utilized and the level of evidence considered. Additionally, it is unclear whether the level of clinical actionability was assessed or evaluated in the study. In particular:
Our response: Evidence for curation and association of drug-gene pairs is confirmed in literature and from our own pharmacogenetic studies on the participants in this cohort. All participants have been genotyped for various pharmacogene variants as illustrated in the different publications in Table 1 further strengthening the evidence used in curation and selection of variants. We have amended the methods section to show this. The method section lines 109-121 now read “ For each cohort, variables assessed included drugs prescribed and comorbidities recorded. Figure 2 shows the workflow of how both pharmacogenes and their variants relevant to Southern African populations were selected. Briefly, to identify the most frequent pharmacogenes relevant to the Southern African patient cohort, all drugs taken by each patient were counted. The top 20 drugs based on prescription frequency were then selected. To select pharmacogenes, we scouted literature for the pharmacology of each drug and identified proteins both transporters and enzymes involved in the disposition of each drug. PharmGKB (www.pharmgkb.org) [11] was used to con-firm the listed genes associated with response to each drug via the pharmacokinetic and pharmacodynamic pathways available in the PharmGKB database. Subsequently, each protein/ enzyme involved in the disposition of the top drugs was noted. Phar-macogenetic variants per gene were subsequently selected based on both prior data from our research group (Table 1), and/ or literature and compared amongst global populations.
Figure 2 Flow chart of both pharmacogene and variant selection. Pharmacogenes were selected after enumeration of the top 20 most frequent co-administered drugs by prescription. Proteins involved in the pharmacodynamics and pharmacokinetics were noted and previous pharmacogenetic studies (Table1 ) were employed to select vari-ants.
All listed studies (Table 1.0) employed participants from our biobank and patient database; including the 3997 described here.”
Table 1: List of studies that have reported on the four cohorts used in the current analysis showing genes that were characterized
|
Cohort |
Reference |
n |
Study population |
Phenotype investigated |
Pharmacogenes genotyped (investigated) |
|
1. Pharmacogenomics of antiretroviral therapy |
Swart et al., 2012 |
282 |
Bantu speaking South Africans |
Efavirenz plasma level, neuropsychological conditions, rashes, hallucination, lack of sleep |
ABCB1 |
|
Swart et al., 2012 |
464 |
Bantu speaking South Africans |
Efavirenz plasma level, neuropsychological conditions, rashes, hallucination, lack of sleep
|
NR1I3, NR1I2 |
|
|
Swart et al., 2013 |
464 |
Bantu speaking South Africans |
Efavirenz plasma level, neuropsychological conditions, rashes, hallucination, lack of sleep
|
CYP1A2, CYP2A6, CYP2B6, CYP3A4, CYP3A5 |
|
|
Kampira et al., 2014 |
203 |
Bantu speaking Malawians |
21% peripheral neuropathy 18% lipodystrophy 17% elevated lactate (.2.5mMol/L) |
mtDNA:nDNA ratio |
|
|
Mpeta et al.; 2016 |
86 |
Banti speaking South Africans |
Lopinavir plasma concentration |
CYP3A4 |
|
|
Carr et al., 2017 |
151 |
Bantu speaking Malawians |
Nevirapine hypersensitivity |
GWAS |
|
|
Carr et al., 2014 |
209 |
Bantu speaking Malawians |
Nevirapine hypersensitivity |
CYP2B6 |
|
|
Mhandire et al., 2015 |
118 |
Bantu speaking Zimbabweans |
Nevirapine hypersensitivity CD4 cell count |
CYP2B6, CYP1A2 |
|
|
2. Atrial fibrillation and mechanical valves (warfarin) |
Ndadza et al., 2021 |
503 |
Black and Mixed ancestry South Africans and Zimbabweans |
Warfarin maintenance dose |
VKORC1, ABACB1, CYP2C9, CYP2C8, CYP1A2, CYP3A4 |
|
Muyambo et al., 2022 |
503 |
Black and Mixed ancestry South Africans and Zimbabweans |
No phenotype investigated |
29 pharmacogenes and 73 variants
|
|
|
3. Pharmacogenetics of tamoxifen |
Unpublished data |
282 |
Black, European and Mixed ancestry South Africans |
Adverse drug reactions reported include hot flushes, pain, blood clots, depression, leg cramps, pins and needles, body aches, stroke, endometrial thickening, endometrial cancer, visual problems and recurrence |
CYP2D6, CYP3A4, CYP3A5, CYP2C9, CYP2C19, CYP2B6, SULT1A1, SULT1E1, SULT2A1, UGT1A4, UGT1A8, UGT1A10, UGT2B7, UGT2B15 |
|
4. PRECODE-hypertension arm |
Unpublished data |
1613 |
Black and Mixed ancestry South Africans |
Resistant hypertension, diabetes, |
ABCB1, CYP3A5, NEDD4L, SCNN1B, CES1, NR3C2, ADRB1 |
|
5. PRECODE-dyslipidemia arm |
Unpublished data |
834 |
Black and Mixed ancestry South Africans |
Statin associated myopathy, diabetes, hypertension |
ABCB1, ABCG2, CYP3A4, CYP3A5, SLCO1B1 |
|
Soko et al., 2016 |
30 |
Bantu speaking Zimbabweans |
Rosuvastatin plasma levels |
ABCG2, SLCO1B1 |
|
|
Soko et al., 2018 |
30 |
Bantu speaking Zimbabweans |
Rosuvastatin plasma levels |
Whole exomes |
Line 111-113: “the author noted that PharmGKB and literature was used to identify genes associated with response to each drug.” It was unclear how was response defined in this context (e.g., efficacy, tolerability, or both?).
Our response: PharmGKB was not used to select variants but to confirm drug metabolism pathways by confirming proteins involved in the pharmacokinetics and disposition of individual drugs. Efficacy of selected genes and variants in response to the drugs are supported by literature and our own research as alluded in the amended methodology. See above comment.
It is important to note that PharmGKB assigns levels of evidence to the annotation of drug-gene pairs. Those with low levels of evidence (e.g., 3 and 4) may not be clinically actionable. The drug-gene pairs included in the evidence-based peer-reviewed guidelines published by CPIC, DPWG, or other professional societies could be informative guidance in addition to PharmGKB.
Our response: PharmGKB was not used to select variants but to confirm drug metabolism pathways by confirming proteins involved in the pharmacokinetics and disposition of individual drugs. The variants were then selected from both literature and our own association studies. The methods section has been amended to read as such. See above comment.
It is also helpful to mention the date of data curation as the database can be periodically updated. For example, the PGx annotations of Amlodipine on PharmGKB at the time of this review are slightly different from what was included in this submission (https://www.pharmgkb.org/chemical/PA448388/clinicalAnnotation)
Our response: Date of access to Pharmgkb is listed in the references in line 480.
#4. Table 2. The rationale for the selection of the variants and their clinical actionability is also unclear to me. For example, some important haplotypes of CYP2C19 are missing, such as *2, *3, *9, *12, *14, *17, etc. Duplication and deletion in CYP2D6 are also missing but very important to consider.
Our response: Polymorphisms mentioned are present on the table 3.
Minor:
#5. To improve the readability, summarizing the clinical and demographic characteristics of study participants in a table could be helpful to improve the readability.
Our response: Demographic and clinical data of study cohorts have been previously reported. Thus line 144 under results now reads “Demographics and clinical characteristics of study participants have been described elsewhere.” References are also provided.
#6. Figure 2 - Confirm the completeness of the top 20 prescribed drugs (study cohort vs population). Some medications mentioned in the text (e.g., lines 132-133: metformin and tramadol) were not shown in the bar plot.
Our response: To allow readability of the figure Figure 2 only lists the top 10 drugs not the entire 20. Line 140 has been corrected to read “Figure 2. Summary of total population showing (a) number of patients reporting condition/disease and (b) top 10 prescribed drugs in this population.”
Please line numbers have moved to accommodate Figure 1 and Table 1.0

Round 2
Reviewer 2 Report
Dear authors, thanks for taking my suggestions into consideration. I think the manuscript has been significantly improved. One minor comment regarding Table 3: Please consider providing star allele designation for CYP2D6 for consistency with other genes. I would also suggest assessing the necessity and feasibility of adding CYP2D6 SNVs.
Author Response
Reviewer’s comments
Dear authors, thanks for taking my suggestions into consideration. I think the manuscript has been significantly improved. One minor comment regarding Table 3: Please consider providing star allele designation for CYP2D6 for consistency with other genes. I would also suggest assessing the necessity and feasibility of adding CYP2D6 SNVs.
Our response
Thank you for your valued review.
Table 3 now contains * nomenclature for all listed CYP2D6 polymorphisms.
Line 397-400 now read “This variant should be considered in clinical utility of pharmacogenomics especially in breast cancer patients. Also to be considered are copy number variants of CYP2D6 such as *2x2 which increases the function of the enzyme and can affect drug metabolism of both tramadol and tamoxifen especially in breast cancer patients within the region.”